# Closed-Loop Controlled Fluid Administration Systems: A Comprehensive Scoping Review

**DOI:** 10.3390/jpm12071168

**Published:** 2022-07-18

**Authors:** Guy Avital, Eric J. Snider, David Berard, Saul J. Vega, Sofia I. Hernandez Torres, Victor A. Convertino, Jose Salinas, Emily N. Boice

**Affiliations:** 1U.S. Army Institute of Surgical Research, JBSA Fort Sam Houston, San Antonio, TX 78234, USA; guy.avital.md.il@gmail.com (G.A.); eric.j.snider3.civ@mail.mil (E.J.S.); david.m.berard3.ctr@mail.mil (D.B.); saul.j.vega.ctr@mail.mil (S.J.V.); sofia.i.hernandeztorres.ctr@mail.mil (S.I.H.T.); victor.a.convertino.civ@mail.mil (V.A.C.); jose.salinas4.civ@mail.mil (J.S.); 2Trauma & Combat Medicine Branch, Surgeon General’s Headquarters, Israel Defense Forces, Ramat-Gan 52620, Israel; 3Division of Anesthesia, Intensive Care & Pain Management, Tel-Aviv Sourasky Medical Center, Tel-Aviv 64239, Israel; 4Battlefield & Health & Trauma Center for Human Integrative Physiology, JBSA Fort Sam Houston, San Antonio, TX 78234, USA; 5Department of Medicine, Uniformed Services University, Bethesda, MD 20814, USA; 6Department of Emergency Medicine, University of Texas Health, San Antonio, TX 78234, USA

**Keywords:** closed loop, decision support, autonomous, automated, controller, artificial intelligence, fluid management, fluid resuscitation, fluid therapy, scoping review

## Abstract

Physiological Closed-Loop Controlled systems continue to take a growing part in clinical practice, offering possibilities of providing more accurate, goal-directed care while reducing clinicians’ cognitive and task load. These systems also provide a standardized approach for the clinical management of the patient, leading to a reduction in care variability across multiple dimensions. For fluid management and administration, the advantages of closed-loop technology are clear, especially in conditions that require precise care to improve outcomes, such as peri-operative care, trauma, and acute burn care. Controller design varies from simplistic to complex designs, based on detailed physiological models and adaptive properties that account for inter-patient and intra-patient variability; their maturity level ranges from theoretical models tested in silico to commercially available, FDA-approved products. This comprehensive scoping review was conducted in order to assess the current technological landscape of this field, describe the systems currently available or under development, and suggest further advancements that may unfold in the coming years. Ten distinct systems were identified and discussed.

## 1. Introduction

Closed-loop controlled systems play a major part in modern life–from thermostats maintaining the room temperature to auto-pilot systems safely keeping airplanes and vehicles en route. These systems are typically designed to maintain a set target value or values, based on a set of system inputs while responding to various environmental changes or other disturbances; thus helping to better manage and automate certain procedures while reducing cognitive and task load [1,2,3].

It is therefore not surprising that closed-loop controllers have an ever-increasing part in clinical care system advancements. Recently, the United States Food and Drug Administration (FDA) had approved a semi-automated closed-loop “artificial pancreas” [4], which aims to improve blood glucose control while decreasing the cognitive and task load in patients, improving their quality of life and glycemic control. In the fields of anesthesia and critical care, closed-loop controlled systems were developed for maintaining ventilation [5], sedation and analgesia [6], vasopressor administration [7], and more. These systems, in the present and the near future, have the potential to unload high-attention, low-complexity tasks from clinicians, freeing cognitive resources for higher-level decision making and formulating the strategy of care for the patient. These systems also carry a significant potential benefit in austere, military and mass casualty scenarios, where the providers often face a high burden of tasks compared to their numbers and expertise.

The systems for the medical closed-loop management of fluids can potentially do all of the above. These systems have the potential for executing more precise fluid regimens in clinical scenarios that require meticulous adherence to therapeutic goals, such as balanced resuscitation as part of the damage-control resuscitation for hemorrhage [8] and fluid management when significant fluid shifts are expected, such as acute burns [9], sepsis [10], and major surgery [11,12].

However, with the medical systems becoming automated and potentially removing the provider entirely from these life-saving interventions, there is a concern that the methods originally developed for the validation of closed-loop controllers for non-medical uses may prove insufficient for the systems required to perform safely in patient-care environments. As a first step towards addressing these concerns, the FDA has conducted a workshop discussing the regulatory implications of Physiological Closed-Loop Controlled (PCLC) medical devices [13], listing concerns such as human factor-related concerns (loss of situational awareness, complacency, skill degradation) and clinical considerations (lack of transparency, sensors’ reliability, handling of disturbances, lack of anticipatory response, knowledge gaps). Despite these concerns, semi-autonomous closed-loop controlled medical systems continue to advance, and the FDA has recently published a draft guidance, Technical Considerations for Medical Devices with Physiologic Closed-Loop Control Technology [14].

PCLC systems are defined by the International Electrotechnical Commission (IEC) as medical equipment systems used to adjust a physiologic variable relative to a command variable (target value) using a feedback variable (measured value) or variables [15]. When applied to fluid management, the closed-loop control systems require a sensor component (e.g., a blood pressure monitor) that senses a feedback variable, a controller, and an actuator (e.g., an infusion pump) that acts to effect the feedback variable. The sensor acts to measure the current value of a parameter of interest (e.g., systolic blood pressure), and this measurement is compared to a target value to determine its error. Based on this error (i.e., the difference between the current measured value and the target value), the controller performs calculations to obtain an actuator setting, and the actuator exerts those settings on the system (e.g., the patient) [16]. As can be seen in some of the systems described in this review, often there will be additional layers of input processing between the sensor and the controller, including the prediction of a future trajectory, so the error will be measured between a clinician-set target and an indirect product of the sensor data, rather than a measured physiological variable. The controller’s decision making can be based on a variety of mechanisms, ranging from simple sets of “if-then” rules, through mathematical calculations based on the error’s trends over time, such as in proportional–integral–derivative (PID) controllers, non-binary rule-based fuzzy logic, up to more advanced methods which apply complex physiological models or population-based predictions to estimate the amount of fluid required to achieve the target value. Another layer that can be included while calculating an actuator setting is adaptivity, in which the patient’s responsiveness to previous actuations is considered.

While a truly autonomous closed-loop controlled system includes all of these components, the semi-autonomous systems may lack direct connectivity to a sensor (requiring the provider to manually input the measured values), an actuator (requiring manual adjustment according to the controller’s output), or both. However, both the sensor and actuator must include a controller component designed to calculate the required actuator adjustment according to the measured values, thus functioning as a decision-support (DS) system. Otherwise, they may have all of the elements of an autonomous system, but still, by design, require the provider’s approval–these are known as “Human-in-loop” systems, and, specifically, “Provider-in-loop” (PIL) systems in medical contexts. A PIL or DS system can potentially be converted to a fully-autonomous one by connecting the automated sensors and/or actuators [13], or by eliminating the requirement for the provider’s approval. The differences between DS, PIL, and fully closed-loop systems are illustrated in Figure 1. The automation level can also be represented by the level of independence the controller has in decision making, as described by Parvinian et al. [13]. It is important to mention that a higher level of automation (LoA) is not necessarily indicative of the level of sophistication, but rather of the existence of additional, supervisory layers that set therapeutic goals for a more long-term approach.

Typically, the closed-loop controlled systems are tested using various methods, with their performance assessed through a variety of criteria, ranging from the engineering criteria from the field of closed-loop controlled systems, such as the Varvel criteria [17], to the criteria measuring clinical outcomes and user satisfaction. Different testing methods include:In silico simulations—These virtual, computer-based tests measure the controller’s ability to respond accurately and robustly in simulated patients’ scenarios. An example for such a testing platform is reported by Bighamian et al. [18]. This method allows for high-throughput tests of the controller at minimal cost and time, using only a computer on existing datasets but cannot test the hardware components;Hardware-in-loop (HIL) testing—A testing method that incorporates hardware components in a physical, manufactured system that simulates a variety of patient scenarios, while measuring the closed-loop controlled system’s performance [19]. These systems can sometimes be merged with an in silico simulation platform [20];In vivo studies (animals)—Testing the entire system’s performance in a real, whole-body physiological system, with subject variability, allows for the additional validation for the system’s performance and enabling measurements of the additional relevant data, such as biochemical markers. It is also an important step towards translation to clinical use. An example of such a study was reported by Marques et al. [21]. An inherent limitation for such studies is the biological difference between testing animals and humans, which not only limits the generalizability of the results, but may undermine the performance of a system that was designed based on human data;Human volunteer trials—These trials have the advantage of testing the system against a true human response, thus overcoming the limitations typical of animal studies. However, ethical constraints may limit the physiological stress that can be imposed on a human volunteer. An example of such a study was reported by Hundeshagen et al. [22];Clinical trials—Testing of a closed-loop controlled system in a clinical setting on real patients can only be completed with a relatively mature system that has been tested by at least some of the methods described above. While it is the ultimate method to prove the system’s real-life benefits, it is more challenging to conduct in emergency scenarios. An example of a clinical trial in a surgical setting was reported by Joosten et al. [23].

The title of “closed-loop controlled fluid administration systems” can describe a variety of systems, varying in rationale, controller type, intended uses, and maturity level. The aim of this comprehensive scoping review was to provide the current landscape of the different systems in various stages of development, to help guide future efforts in this evolving field of research. The review was limited to systems that were designed as a feedback loop, re-assessing the patient’s condition and adjusting its output accordingly. The decision support systems, as long as they contained a feedback loop (meaning these systems respond to repetitive input from the patient, either directly from sensors or through the provider) were included in this review. The inclusion of certain decision support systems that mimic closed-loop control in their operation acknowledges that the difference between an autonomous closed-loop controlled system and a semi-autonomous decision support system was in the hardware components’ connectivity, while the controller algorithms can be virtually similar.

A preliminary search of PubMed, the Cochrane Database of Systematic Reviews, and *JBI Evidence Synthesis* was conducted and no current or underway systematic reviews or scoping reviews on the topic were identified. While some of the manuscripts offered a narrative on closed-loop fluid administrations systems [16,24], none of them provided a systematic, comprehensive view of the state-of-the-art and current progress in that field.

## 2. Materials and Methods

### 2.1. Registration and Protocol

This scoping review was conducted in accordance with the JBI methodology for scoping reviews [25], and the manuscript was composed in accordance with the adapted version of the Preferred Reporting Items for Systematic Reviews and Meta-analyses extension for scoping review (PRISMA-ScR) guidelines [26]. The review protocol was pre-uploaded and displayed on the Open Science Framework (https://osf.io/nr7th/?view_only=91b33c2bad6545ee8f4fe8ad8570e364) on 15 January 2022, with no subsequent edits.

### 2.2. Eligibility Criteria

This review aimed to identify all of the closed-loop controlled systems for the management of fluid therapy in patients. Fully autonomous, semi-autonomous (provider-in-loop) and decision-support systems were included, as long as their logic included a complete closed loop, meaning the patient input was continuously or frequently re-assessed and output was adjusted based on this input. Only the systems with peer-reviewed publications were included. No exclusion was made with regard to the patient population. Only the English language reports were sought.

### 2.3. Search Strategy and Information Sources

To identify the potentially relevant documents, PubMed and Scopus were searched for publications from 1 January 2012 to 19 January 2022 (day of the search), as systems not reported on for over 10 years were unlikely to be continued with future clinical development. Search strategy is described in the Appendix A and was drafted by the authors and refined by a series of preliminary searches, with consultation from an experienced librarian. In addition, patent records were searched in Scopus, which incorporates several patenting offices globally, including the United States Patent and Trademark Office (USPTO), European Patent Office (EPO), and others. The patent search was conducted with a time limitation. The search results were imported to a spreadsheet (Excel, Microsoft Corp., Redmond, WA, USA), and the duplicates were removed. The identified patents were compared with identified manuscripts based on authors’ names. In the case of patents where the matching manuscripts were not found, PubMed was searched for the first patent holder’s name in an attempt to identify the manuscripts that were missed by the search strategy. The eligible manuscripts were also searched for relevant references by the reviewing researchers. The authors of the manuscripts on relevant systems were contacted via e-mail and/or virtual meeting to fill data gaps if present.

### 2.4. Selection Process

Following the search, all of the identified abstracts were collated and uploaded into a spreadsheet and the duplicates were removed. The titles and abstracts were then independently screened by two reviewers, an engineer and a clinician, for assessment against the inclusion criteria for the review. The relevant abstracts were grouped according to author groups and full texts were retrieved. In case a relevant citation was located in a publication’s reference list, it was retrieved and reviewed as well. This was performed by a single reviewer (an engineer), and re-examined by a second reviewer (a clinician). Any disagreements that arose between the reviewers at each stage of the selection and the data extraction processes were resolved through discussion, based on the pre-determined scope of the review. The patent abstracts were screened directly on the Scopus website by a single reviewer, and only the potentially relevant patents were included in the spreadsheet, as an import of all of the patent search results was not technically possible. To reduce the risk of missing patents, a very inclusive approach was taken, later narrowed down by a second reviewer. The relevant patent abstracts were matched to the identified groups, and, in case no match was found, an additional PubMed search based on the patent holders’ names was conducted. In addition, another PubMed search of the names of the first and last authors of the most recent publication on each identified system, as well as the system’s name if available, was conducted to secure the most updated data. All of the newly identified relevant manuscripts were added to the spreadsheet. The full texts of all of the manuscripts within each group were assessed in detail by two reviewers, an engineer and a clinician, for eligibility, and the pre-determined data were extracted to a second spreadsheet, i.e., the extraction tool. No critical appraisal of the selected reports was performed. The results of the search and the study inclusion process are presented in an adapted version of the Preferred Reporting Items for Systematic Reviews and Meta-analyses extension for scoping review (PRISMA-ScR) flow diagram (Figure 2) [26].

### 2.5. Deviations from Original Protocol

As part of the iterative nature of scoping reviews, and in order to provide the most comprehensive and current report during the process of data gathering and processing, the following adaptations to the original protocol were made:As the patent registries often include future possible uses of the described patent, references to “closed-loop systems” are often not indicative of the actual existence of such systems. Therefore, identified patent registries were used as a basis for a search of peer-reviewed publications by the first patent holder’s name in PubMed, but were not considered sufficient to _describe_ a system by themselves;In order to confirm that the most updated version of each system is described, an additional PubMed search was conducted, based on the first and last authors’ name of the latest manuscript found on the system. The system’s name, if described, was also included in a search, in addition to communication with the authors themselves;Some of the manuscripts, which described experiments with closed-loop controlled systems, were excluded at the final step after communication with the authors clarified that further development of the systems would not be pursued., As such, the aim of this report was to provide a landscape of the foreseeable future of closed-loop fluid management;While the original strategy was to review the manuscripts on each system in a new-to-old fashion until all of the required data were obtained, we decided to review all of the relevant manuscripts to deepen our understanding on each system.

### 2.6. Data Extraction

The data were extracted from the manuscripts included in the scoping review by two reviewers, using the data extraction tool developed by the reviewers. The extracted data included all of the specific details about the properties of the different systems, in a way that allowed for a qualitative comparison between them. The extracted data items included: publication year; degree of automation (decision support/provider-in-loop/closed loop); level of automation (as described by Parvinian et al. [13], one-completely manual control, two-target set by provider, three-target set by system, four-full automation, including type of therapy and decision to initiate or cease therapy); rationale; real-time adaptivity; controller type; inputs; outputs; optimization goals; intended use case; fluids used; stage of research; regulatory status; testing platforms; performance specifications; metrics provided; and source of funding for the studies. In addition, notes were taken of the significant properties of the systems that were deemed worth mentioning.

### 2.7. Synthesis of Results

The various reports of each system were used to summarize extracted data items from the various reports to a single answer per item per system, and displayed in a single table. The newer data were preferred over old data. Some of the gaps were filled by authors’ responses, although published data were prioritized. Some of the data items were merged to make the data more accessible, while items that were found to be less informative were removed. Additional interesting comments on each system were collated, to be displayed as free text.

## 3. Results

### 3.1. Source Selection

The search, filtering and review process is presented in an adaptation of the PRISMA-ScR flow diagram (Figure 2). Following removal of the duplicates, 307 potentially relevant abstracts were identified. Following their review, 63 were deemed eligible for full manuscript review, out of which 46 met the criteria to be used for data extraction. A total of 5178 patent registries were identified, out of which 129 were possibly describing closed loop fluid management systems. Out of those, 45 were describing systems already identified by the manuscript search, and the results from searching the first patent-holder name led to the discovery of four additional manuscripts, which, upon revision, were all found not to describe a closed-loop controlled system. A search based on the first and last authors’ and systems’ names yielded nine more manuscripts, a manual review of the reference lists yielded three more, and the authors provided one additional relevant manuscript. The authors’ responses were sought and not received for two of the systems. Additional manuscripts were found that did not meet the criteria for inclusion, but were helpful for the purpose of broadening of the discussion.

Due to the scope of this review, systems that were meant for vasopressor infusion control were not included, and only the fluid management component was addressed in the cases where both fluid management and vasopressor control were described. Two exceptions were the systems described by Libert et al. [27,28] and by Markevicius et al. [29,30], where these two components were essentially inseparable.

### 3.2. Synthesis of Results

The results of the data extraction are detailed in Table 1. A reference for the sources for the information on each system can be found under the category “sources”. All of the funding sources described in any of the sources are detailed under the category “funding”.

None of the systems had a level of automation of three or above, meaning they were either decision support systems or level two systems, with the goal value being manually set and not re-adjusted by the system.

The following is a brief description of each of the various systems identified by our review methodology:

#### 3.2.1. Acumen-Assisted Fluid Management (AFM)

This system was originally designed as a completely closed-loop system for automating the peri-operative Goal-Directed Fluid Therapy (GDFT) for the improvement of clinical outcomes after major surgery. Its goal was to optimize the patient’s place on the Frank–Starling curve, expressed by *n* scaled value of the percent increase in stroke volume (SV) in response to a 500 mL fluid bolus. This value is not directly measured, but rather predicted based on a comparison with experimental population data. It is then modified by an adaptive factor derived from previous measure responses from the patient, and a recommendation for fluid administration is provided by a rule-based engine. It was tested in various platforms, including several clinical trials, and was purchased by Edwards Lifesciences (Irvine, CA, USA) in 2014. This system was commercialized under the name “Acumen AFM” as a decision support software. As this system is meant for GDFT, its response time and algorithm are not designed for extreme situations, such as resuscitation from hemorrhagic shock.

#### 3.2.2. Burn Navigator

This commercial product by Arcos Medical (Missouri City, TX, USA) is derived from the Burn Resuscitation Decision Support System (BRDSS) developed by the US Army Institute of Surgical Research (USAISR). It is meant to optimize major burn (>20% Total Body Surface Area) casualties’ fluid resuscitation in the first 24 to 48 h. A population-based formula predicts next hour urine output (UO) from patient demographics, burn data and UO in the last 3 h. A second formula predicts the UO response to fluid administration based on a population-based model, and a recommendation for alteration of the hourly infusion rate is issued and adjusted by a series of practical “business rules” before issuing a final recommendation, with the aim of maintaining UO in the 30–50 mL/h range. The use of predicted UO, rather than last measured UO, helps to bridge the long response time resulting from measurement of UO on an hourly basis and “stay ahead” of the changes, instead of responding late. A fully closed loop system was also developed and prototyped for research and animal testing, using a commercial urinometer system as the input to control an (non-FDA cleared) infusion pump. However, this system has not been commercialized.

#### 3.2.3. University of Maryland (UMD) Systems

The team from UMD has published on two prominent closed-loop fluid administration systems intended for hemorrhagic shock resuscitation.

The first one, referred to in this review as “UMD 2016”, is still in early phases of development. It is meant for accurate fluid resuscitation through assessment of blood volume deficiency via a complex physiological model linking arterial waveform and electrocardiographic response to volume input with fractional blood volume. Then, it predicts a response of end-diastolic volume to future volume infusion, using a complex compartmental model, an adaptive module and a proportional–integral–derivative (PID) controller. According to communication with the authors, future plans for this system include some alterations of the model, and eventual transfer to a decision support system.

The second one, referred to as “CAC” in this review, is meant for similar purposes, but using a different rationale. The original version of this controller, named “Model reference adaptive control” (MRAC), used a physiological model-based estimation of blood volume from the non-invasive measurement of the hemoglobin concentration’s response to fluid bolus, which requires an initial bolus phase prior to the initiation of the feedback loop. An adaptive layer corrects for inter-patient as well as intra-patient (thanks to its recursive action) variability, making the adaptive layer the part of the closed-loop system that responds to feedback from the sensors. Later iterations integrated the use of blood pressure as the controlled variable (by expanding the model to account for blood volume–blood pressure relationship), with combined feedback of the blood volume estimation and blood pressure, also known as “Composite adaptive control (CAC) with blood pressure and hematocrit feedback” as the optimal version of the system, based on in silico performance evaluations focused on Varvel’s criteria.

#### 3.2.4. Tübingen

This system, developed at the University of Tübingen, Germany, was presented as part of an array of PCLC systems meant for the maintenance of a critical patient, simulated by a healthy, unchallenged swine model, with the goal of observing the co-performance of the systems. This fluid management system provides an assessment of volume status, based on the response of systolic pressure to a 20 s inspiratory pause. A value called the VNA delta (VNA-Volume Needed Analysis) is calculated as the difference between the maximal and minimal systolic blood pressure measurements during the maneuver. In case the VNA delta is above a set threshold, a small fluid bolus is administered and the sampling interval is decreased from 60 to 15 min. The response times vary from 15 to 60 min. Although this system was meant for fluid maintenance, it is not suited to treat cases of severe hypovolemia, such as hemorrhagic shock. According to our communication with the authors, the focus of development is on continued basic research for the integration of multiple PCLCs into critical care scenarios, rather than pursuit of commercialization.

#### 3.2.5. Renal Guard

Probably the simplest of all of the identified PCLC systems, its controller is comprised of a single rule: fluid infusion rate equals UO. With the addition of a diuretic, this system can lead to a high UO and a short transit time for solutes in the renal tubules. The main aim of this system, evaluated in various conflicting studies, is the prevention of contrast media-induced nephropathy. Other indications, such as rhabdomyolysis, were also reported. Although included in this report for the sake of comprehensiveness, it does not completely meet the criteria for a CL system, as it is not aimed at optimizing a physiological value to a set target. However, this system actuates the infusion rate based on a measured physiological value in a CL fashion.

#### 3.2.6. Trauma Tab

Reported by Libert et al. from Université Paris-Sud, this system is aimed at balancing crystalloid and vasopressor infusion for the maintenance of permissive hypotension, based on experiments involving profound, prolonged, hemorrhagic shock in small and large animal models. The function of the Trauma Tab system incorporates a set of “phases”, with a fuzzy logic controller that imitates clinical decision making, rather than on a physiological model or population-derived data. Their studies also demonstrate a simpler, proportional–integral (PI) controller. The results are reported in both engineering (Varvel’s criteria) and clinical terms. Each study protocol examines several controller configurations, and the future plans for this system are not discussed in the reports. Despite reaching out to the authors, no response was received before publication of this review.

#### 3.2.7. E-Fusion

Developed by a private company named Autonomous Healthcare, the E-Fusion system was initially reported on by Gholami et al., based on data collected using a canine model. This system is based on a bi-compartmental model of fluid dynamics, adapted to the specific patient through neural networks’ techniques and is meant to optimize stroke volume variation (SVV) as an indicator of preload. According to the authors, the system is in testing phases and is not yet commercially available.

#### 3.2.8. Semi-Closed Loop (SCL) Infusion System

Reported on by Markevicius et al., this system aims at optimizing GDFT by setting a target MAP and infusing fluids up to a point of imminent edema, as diagnosed using a minimal Volume Loading Test (mVLT) method. In this method, small boluses are infused and the resulting hemodilution is measured. The absence of hemodilution is used as an indication of imminent edema, in which case fluid administration is stopped and blood pressure is maintained by a closed-loop vasopressor titration controller, which is constantly working in the background. In addition, the system provides a recommendation for the infusion of packed red blood cells (PRBC) when hemoglobin concentration falls below a critical threshold. All of the above actions are governed by a rule-based engine, designed to imitate clinical decision making in which the target MAP is achieved by maximizing preload while avoiding edema, and relying on vasopressors to bridge any remaining gap between the measured and target MAP. While the logic of the system is explained in reviewed manuscripts, the results of the in silico trials are not detailed. Despite reaching out to the authors, no response was received before publication of this review.

#### 3.2.9. Adaptive Resuscitation Controller (ARC)

ARC was also developed at the US Army Institute of Surgical Research, and uses experimental animal data from previous porcine hemorrhage models to predict the pressure response to infused volume. The controller’s algorithm contains an adaptive correction factor that correlates predictions based on the previous responses of the current patient, and a proportional component to regulate the infusion rate, so that the further the MAP is from the setpoint, the faster the volume will be infused. This controller was tested in a hardware-in-loop platform, evaluating the pressure responses to two types of simulated fluids (whole blood and crystalloid). Further development of the ARC is targeted for its optimization for severe hemorrhage animal models, seen in combat casualty care, and its role as part of a larger supervisory algorithm, designed to manage additional controllers and their interactions.

## 4. Discussion

In this scoping review, we have identified ten distinct closed-loop controlled patient-fluid management systems, at various maturity levels, for multiple intended-use cases.

Several general observations can be drawn from the results of this review, the first one being the multitude of different approaches taken for control logic. These differences stem, first and foremost, from the different intended clinical use cases—from the simplicity of forced diuresis fostered by RenalGuard, through the relevant simplicity of the MAP-goal directed resuscitation of the acute hemorrhagic patient, and ending with the complex fluid dynamics of patients with extensive burns, major surgery or prolonged intensive care.

With the exception of the simplest systems, most of the systems use one of two approaches to predict the responses based on patient sensor input:A model-based approach, in which predictions are made based on mathematical models aimed to mimic real physiological processes, most commonly compartmental models, describing the fluid balance between body compartments;A population-based approach, in which empirical data collected from clinical patient records or experiments (either human or animal) are used to create prediction formulas.

Another observation concerns the sensor and controller sampling rate, which is an important factor in the consideration of the PCLC system development—continuous or frequent measurements decrease the effect of a controller’s erroneous calculation through rapid re-calculation. Conversely, infrequent inputs, such as urine output, necessitate accurate calculations as well as predictions of future trends (such as in the case of the Burn Navigator), so that the calculations are based on data as current as possible, so the system does not lag behind the patient.

Third, adaptivity is described differently across reports. The most useful test for adaptivity would probably be whether the system will respond differently to identical momentary inputs from the patient, based on previous responses by the same patient, effectively “learning” the patient. This trait is crucial for the accuracy of both the model-based and population-based algorithms, as it corrects for inter-patient (and, over time, intra-patient) variability.

Last, the ability of any PCLC medical device to provide accurate control of goal-directed resuscitation is dependent on the use of physiological signals that reflect a feedback target with the greatest sensitivity and specificity. Another tension exists between the desire to increase the system’s usability by incorporating non-invasive sensors and the requirement for high-fidelity physiological input. MAP has been used traditionally by the clinical community as the standard feedback signal for manual resuscitation, despite the existence of data that indicate its relatively low sensitivity in time (i.e., significant delay in change) [77,78]. In this regard, compelling evidence has emerged over the last decade that arterial waveform feature analysis (AWFA) provides greater sensitivity and specificity for the measurement of the circulatory status of individual patients, when compared to MAP and other standard vital signs.

An important hurdle concerning regulation and implementation is the algorithm complexity and transparency—a simple physiological logic is more understandable to the clinician, promoting better understanding of the system’s operation and shortcomings [13], hence increasing safety, which in turn facilitates regulatory approval.

The advanced testing platforms can often be seen as an indicator of system maturity, as systems tested in clinical settings or with animal experiments are generally considered to be at a higher readiness level. Nevertheless, testing in advanced platforms does not guarantee a high maturity level, as the performance of multitudes of robustness assessments, which requires high throughput testing platforms (e.g., in silico or HIL), is essential for true system maturity and reliability.

While significant effort was made to find and describe the entire technological landscape of closed-loop resuscitation systems, some limitations to our study exist. It is possible that there are other relevant systems, as well as later reports of described systems, that were not found using our search protocol, especially when no response to our inquiries was received from the authors. The lack of critical appraisal, which was designed to make this review as inclusive as possible, may have led to the inclusion of systems with questionable design and testing methods. We attempted to reduce this risk by limiting this report to rely on peer-reviewed publications, yet we encourage the reader to further conduct their own investigation about the described systems with a robust critical approach. Most importantly, such a comprehensive review cannot do justice to the complexity and ingenuity of each of the described systems, and should only be seen as an introduction to this field for the curious reader.

## 5. Future Directions

With regard to future directions in this field, there are several trends that might be expected:Further maturation of the described systems as well as introduction of new ones;Increased adoption of closed-loop controlled fluid administration. The first scenario, where these systems can be safely operated, will probably be the operating room, where constant supervision by an anesthesiologist provides an important safety net;Deeper understanding of fluid dynamics and their translation to ever-more-complex computational models, meant for better accuracy and validity of both controllers and in silico testing platforms [18,79,80];Introduction of new modalities of artificial intelligence, such as reinforcement learning [81,82] and other deep-learning modalities. While there’s increasing use of deep-learning for anesthesia and critical care-related applications [83,84], we have not identified detailed reports on deep-learning-based systems matching our inclusion criteria, meaning we have not identified a system that incorporates deep-learning-based capabilities into a CL system (or, for that matter, a DS system with a feedback loop of repeat evaluations);Continuing formation of a regulatory pipeline dedicated to autonomous and semi-autonomous controlled systems;Increased use of non-invasive sensors in closed-loop fluid administration systems, as their reliability will gradually increase [43,85], as well as artificial intelligence-based advanced sensing modalities (specifically, feature extraction), such as arterial waveform feature analysis [77,78], aimed at providing personalized resuscitation goals;Gradual increase in the degree of automation—from a regulatory standpoint, decision support systems are generally considered safer and easier to approve. However, they do not offer the same mental offloading and adherence as a closed-loop or even a provider-in-loop system can potentially offer. These advancements require, other than regulatory endorsement, the integration of reliable sensors and actuators (i.e., infusion pumps), so that controllers’ commands will be based on accurate data and executed precisely to ensure patient safety;Increase in the level of automation—as more life-support systems are being automated, supervisory controllers will be required to integrate them to accommodate the physiological interaction between the body systems. These controllers will adjust the target goals for the sub-controllers (e.g., more permissive fluid resuscitation to accommodate for the need to increase positive end-expiratory pressure), bringing composite systems to LoAs of three or four.

Most importantly, the closed-loop systems are not meant to replace clinicians, and will not do so anytime in the foreseeable future. Instead, they are designed to provide the clinician freedom from the burdens imposed by technical, distracting and wearisome tasks, allowing for more accurate performance and allowing the clinician to focus on making the right important decisions for the benefit of improved patient outcomes.

## Figures and Tables

**Figure 1 jpm-12-01168-f001:**
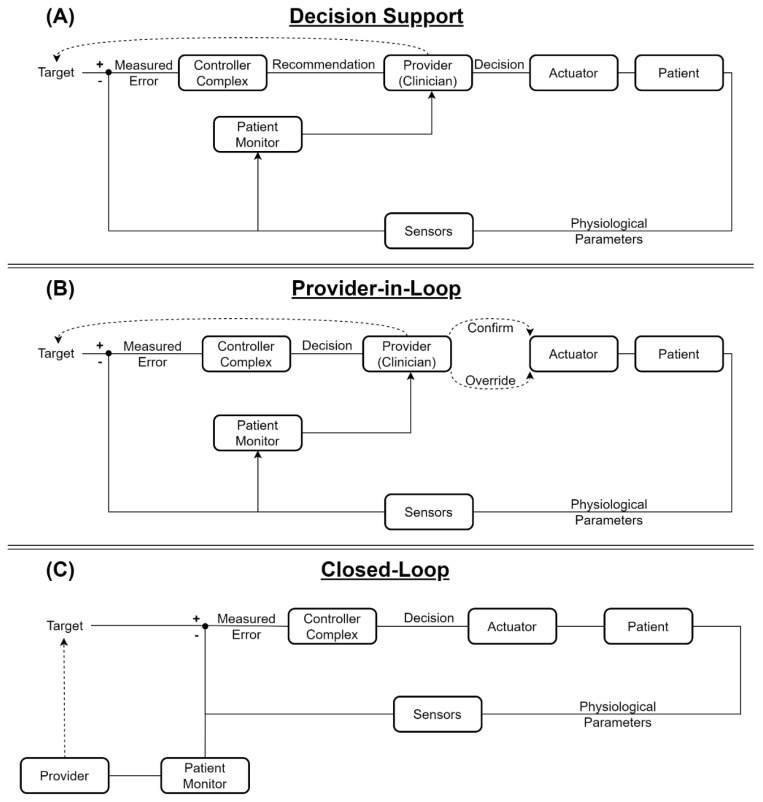
Flow diagrams describing physiological (**A**) Decision Support; (**B**) Provider-in-Loop; and (**C**) Closed-Loop controlled systems.

**Figure 2 jpm-12-01168-f002:**
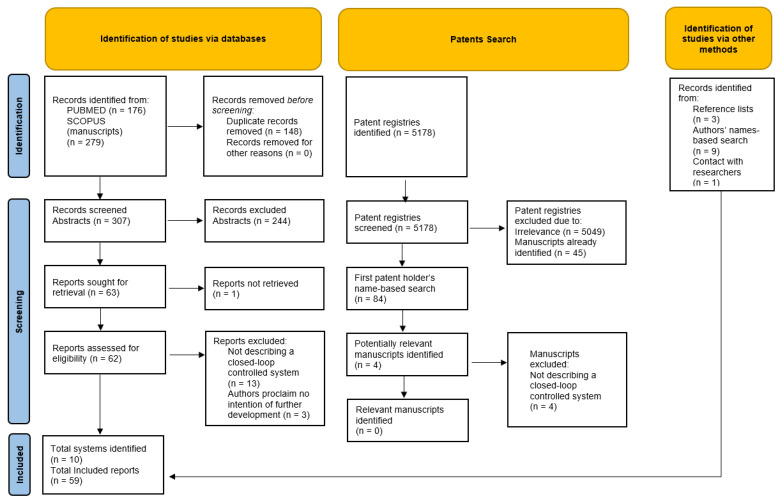
PRISMA-ScR Flow diagram of the scoping review format.

**Table 1 jpm-12-01168-t001:** Identified closed-loop controlled systems for fluid resuscitation.

System	AFM	Burn Navigator	UMD 2016	CAC	Tubingen	RenalGuard	TraumaTab	E-Fusion	SCL Infusion System	ARC
Year last reported	2021	2021	2016	2022	2018	2022	2021	2021	2014	2022
Degree of automation	DS **	DS	CL	CL	CL	CL	CL	CL	PIL	CL
Rationale	Population based	Population based	Model Based	Model Based	Pmcf Method	No prediction	Fuzzy Logic and “phase recognition”	Model Based	Rule based	Population based
Adaptivity	Yes	Yes	Yes	Yes	Partial ***	No	Yes	Yes	No	Yes
Inputs *	HR, MAP, SV, SVV	UO	Arterial waveform and ECG	MAP, SpHb	SBP (VNA)	UO	SBP	SVV	MAP, SpHb	MAP
Output	Suggestion for fluid administration	Recommended fluid rate	Infusion volume	Infusion volume	2 mL/kg bolus	Infusion rate	Infusion rate	Infusion rate	Recommendations on fluid rate, vasopressor titration and PRBC administration	Infusion rate
Optimization goal	% Increase in SV following fluid bolus	30 < UO < 50	% Increase in EDV following fluid bolus	MAP	VNA ≤ 10	Infusion rate = UO	SBP	SVV ≤ 13	MAP, SpHb	Target MAP Value
Intended use case	GDFT for peri-operative care	Acute burn resuscitation	Hemorrhagic shock resuscitation	Hemorrhagic shock resuscitation	ICU Patient Maintenance	Forced Diuresis	Hemorrhagic shock resuscitation	Peri-operative and trauma care	Peri-operative care	Hemorrhagic shock resuscitation
Fluids used or simulated	Crystalloids/colloids/blood	Ringer’s Lactate	Crystalloids	Crystalloids	Crystalloids	Normal Saline	Normal Saline + Norepinephrine	Ringer’s Lactate	Crystalloids, PRBC, Adrenaline	Crystalloids/whole blood
Most advanced research stage	Clinical trials	Clinical trials	In silico testing	In silico testing	Large Animal Pilot Study	Clinical Trials	Large Animal pilot study	Large Animal Pilot Study	In silico testing	Hardware-in-loop
Regulatory Status	FDA and CE approved	FDA Approved	N/A	N/A	Unknown	CE approved, pending FDA approval	Unknown	Unknown	Unknown	N/A
Performance metrics provided	Agreement of user with recommendations, effectiveness of recommended boluses comparing to user-initiated	Clinical outcomes, Users’ satisfaction	Algorithm’s prediction accuracy	Varvel’s criteria	% of time spent under VNA delta threshold	Difference between measured UO and infused volume, clinical outcomes	Varvel’s criteria, clinical markers	Time to target, fluid balance	Not specified	Time to target, Fluid balance
Funding sources disclosed in studies	Edwards Lifesciences, NIH, ESIC, Brugmann Foundation	US DoD, NIH	US-ONR	Fulbright program, US-NSF, US-ONR	Institutional funding from B. Braun	NIHR, RenalGuard solutions, PLC Medical	French Ministry of Defense	Autonomous Health Inc.	European Union	US DoD
Sources	[23,24,31,32,33,34,35,36,37,38,39,40,41,42,43,44,45,46,47,48,49,50,51,52]	[53,54,55,56,57,58,59,60,61,62]	[63]	[64,65]	[66]	[67,68,69,70,71,72,73,74]	[27,28]	[16,75]	[29,30]	[76]

* Refers to variables monitored repeatedly, not only when initiating the system (e.g., patient demographics). ** Based on a tested CL system. *** Only sampling rate is adapted in response to input. Acronyms: AFM-Automated Fluid Management; DS-Decision Support; HR-Heart rate; MAP-Mean arterial pressure; SV-Stroke volume; SVV-Stroke Volume Variation; GDFT-Goal directed fluid therapy; FDA-Food and Drug Administration; CE-Conformité Européenne; NIH-National Institute of Health; ESIC-European Society of Intensive Care; UO-Urine output; DoD-Department of Defense; UMD-University of Maryland; CL-Closed-loop; ECG-Electrocardiogram; EDV-End diastolic volume; US-ONR-United States Office of Naval Research; CAC-Composite Adaptive Control; SpHb-Blood Hemoglobin Concentration; US-NSF-United States National Science Foundation; SBP-Systolic Blood Pressure; Pmcf-Mean circulatory filling pressure; VNA-Volume Needed Analysis; ICU-Intensive Care Unit; NIHR-National Institutes of Health and Care Research; SCL-Semi-closed loop; PRBC-Packed red blood cells; PPV-Pulse Pressure Variation; PVI-Plethysmography Variability Index; ARC-Adaptive resuscitation controller.

## Data Availability

The datasets generated during and/or analyzed during the current study are available from the corresponding author upon reasonable request.

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
