# Peer review of "Closed-Loop Controlled Fluid Administration Systems: A Comprehensive Scoping Review"

_jpm, 2022, doi:10.3390/jpm12071168_

Round 1
Reviewer 1 Report
The article is a comprehensive scoping review of closed-loop control fluid administration systems. To be publishable the following revisions are required.
1. The graphical explanation of the article is too simple and not informative, it is expected the authors give a better graphical representation of the work. Like Fig 1 that is too general and too simplified.
2. The term artificial intelligence is one of the article keywords is almost not reviewed in the article, in artificial intelligence, deep learning is in use in many controlling systems, as in given following works, it is worth mentioning them; ``A Deep Learning-Based Approach for Generation Expansion Planning Considering Power Plants Lifetime''(2021), ``Machine Learning, Deep Learning, and Closed Loop Devices—Anesthesia Delivery''(2021), ``Artificial intelligence and anesthesia: A narrative review''(2022).
3. The section of Future direction should be merged with a conclusion.
Author Response
- The graphical explanation of the article is too simple and not informative, it is expected the authors give a better graphical representation of the work. Like Fig 1 that is too general and too simplified.
While the original idea was to provide a basic concept of what consisted in a PCLC system, we have taken this opportunity to improve it by using it to demonstrate the differences between decision support, provider-in-loop and fully closed-loop system.
Fig. 2 is directly derived from the PRISMA-ScR guidelines, and we feel obligated to leave it unchanged. We believe that due the nature of a scoping review, it is best represented by display in the form of a comparative table like Table 1. We believe that with this modification to Figure 1, complimented with Figure 2 and Table 1, the graphical explanations nicely supplement the text.
- The term artificial intelligence is one of the article keywords is almost not reviewed in the article, in artificial intelligence, deep learning is in use in many controlling systems, as in given following works, it is worth mentioning them; ``A Deep Learning-Based Approach for Generation Expansion Planning Considering Power Plants Lifetime''(2021), ``Machine Learning, Deep Learning, and Closed Loop Devices—Anesthesia Delivery''(2021), ``Artificial intelligence and anesthesia: A narrative review''(2022).
We thank the reviewer for their insightful observation. As correctly stated, AI and specifically deep learning are playing an ever-increasing role in medical care such as anesthesia. However, our review process did not identify any deep-learning driven closed-loop fluid management system. We did mention this subject as part of the Future Directions section, and based on the reviewer’s comment have made this mention more prominent, together with referencing part of the above mentioned manuscripts (Lines 530-536).
- The section of Future direction should be merged with a conclusion.
We thank the reviewer for this idea. Our previous work has typically included a summary statement in the form of a dedicated Conclusion section. As scoping reviews are unique in process and format, we believe we aided accessibility and ease of reading of the manuscript by focusing most of the attention to the individual systems in the Results section and how those systems compare in the Discussion. Closed-loop controllers are a continuously evolving field of research, and a dedicated conclusion section did not seem worth the valuable real estate in the manuscript. We chose instead to offer the readers an exciting glimpse at the next steps of the field in the Future Directions section.
Reviewer 2 Report
Thanks the authors for the opportunity to review such an intresting manuscript.
Very well organized work that analyzes in a comprehensive way the theme object of the title.
Comments Updated on 13 July 2022:
Closed-loop controlled systems play a growing part in clinical practice; authors try to describe the entire technological landscape of closed-loop systems.
This topic, although still little known, will be of great relevance in the near future. For this reason I consider this review very useful to disseminate the usefulness and technological advancement of these devices. The review cover all clinical aspects where this technology is currently already in use (diabetes treatment, infusional therapy, etc.)
The authors present the various products in use, the logic of operation and the future prospects of this technology.
The paper is well written and authors addressed the question posed. the bibliography is adequate for the purpose.
There are several reviews but many are specific to one aspect of the use of this technology. This one under review is the most extensive and complete
Author Response
- Thanks the authors for the opportunity to review such an interesting manuscript. Very well organized work that analyzes in a comprehensive way, the theme object of the title.
We appreciate the reviewer taking the time to thoroughly review the manuscript.
Reviewer 3 Report
The authors have put together a very comprehensive, thorough, in-depth review of Closed-Loop Controlled Fluid Administration Systems.
- I appreciated the background given in the introduction to basics and closed-loop systems, as well as background and update of FDA regulations.
- Treatment of topics regarding modulations is helpful. Bullet pointed descriptions of ways to test systems is also useful for readers
- The systematic approach to their search, with use of clinicians and engineers and multiple levels of reviewers was noted
- All figures and tables are appropriate and helpful for readers
- Discussion section could be improved. I think starting with a clear summary of their findings. Then what the limitations and barriers to progress are. These things are discussed but I think a bit more clarity here would help readers.
Author Response
- Discussion section could be improved. I think starting with a clear summary of their findings. Then what the limitations and barriers to progress are. These things are discussed but I think a bit more clarity here would help readers.
We thank the reviewer for this feedback and thoughtful suggestion. To address this, we have made some changes to the structure of the discussion section, including changing the order of some of the paragraphs and itemizing the observations drawn from the results, and the addition of a brief mention of the results, in order to avoid redundancy. We hope and believe that these changes will make the discussion section more accessible and can be seen at Lines 447 – 479.